# Oxidation–Responsive Emulsions Stabilized with Poly(Vinyl Pyrrolidone-*co*-allyl Phenyl Sulfide)

**DOI:** 10.3390/polym12020498

**Published:** 2020-02-24

**Authors:** Seok Ho Park, Jin-Chul Kim

**Affiliations:** Department of Medical Biomaterials Engineering, College of Biomedical Science and Institute of Bioscience and Biotechnology, Kangwon National University, 192-1, Hyoja 2 dong, Chuncheon, Kangwon-do 200-701, Korea; sukho1002@kangwon.ac.kr

**Keywords:** emulsion, oxidizable copolymers, allyl phenyl sulfide, demulsification

## Abstract

Oxidation-responsive emulsions were obtained by stabilizing mineral oil droplets using amphiphilic poly(vinyl pyrrolidone-co-allyl phenyl sulfide) (P(VP-APS)). ^1^H nuclear magnetic resonance (NMR) spectroscopy revealed that P(VP-APS) whose APS content was 0%, 3.28%, 3.43% and 4.58% were successfully prepared by free radical reaction and the sulfide of APS was oxidized by H_2_O_2_ treatment. X-ray Photoelectron Spectroscopy (XPS) also disclosed that the sulfide of APS was oxidized to sulfone by the oxidizing agent. The optical density of copolymer solutions and the interfacial activity of the copolymers markedly decreased by H_2_O_2_ treatment possibly because the sulfide of APS was oxidized and the amphiphilicity of the copolymers were weakened. The increase rate of the oil droplet diameter of the emulsions was outstandingly promoted when H_2_O_2_ solution (10%, v/v) was used as an aqueous phase. The phase separation of the emulsions was also expedited by the oxidizing agent. The oxidation of APS and the weakened interfacial activity were thought to be a main reason for the demulsification of P(VP-APS)-stabilized emulsions.

## 1. Introduction

Surface-active molecules are able to stabilize oil or water droplets in the continuous phase of emulsions by reducing the oil/water interfacial energy. If the emulsifiers change their molecular conformation in answer to stimuli including light irradiation, temperature change and pH value change, the droplets would undergo coalescence, leading to the phase separation when the emulsions are exposed to the stimuli [1,2,3,4,5,6]. Emulsions are need to be stabilized with the stimuli-responsive emulsifiers, if they are required to be demulsified readily on demand. In order to prepare temperature and pH-responsive emulsions, poly(*N*-isopropylacrylamide-*co*-methacrylic acid-*co*-octadecyl acrylate) was used as an emulsifier [7]. Upon change in the temperature and the pH value of medium, the copolymer would take a contracted form and not fully cover the O/W interface, resulting in the coalescence and the phase separation. A temperature- and photo- responsive emulsion was prepared using poly(2-hydroxyethyl acrylate-*co*-coumaryl acrylate-*co*-2-ethylhexyl acrylate) [8]. The copolymer exhibited lower critical solution temperature (LCST) in an aqueous solution, thus it would be condensed a contracted form when the temperature of medium was raised above the phase transition temperature. When the emulsion was heated up above LCST, the area the copolymer chains could cover would hardly be large enough to stabilize the O/W interface. The copolymer chains would also be condensed when they were subjected to UV irradiation, because coumaryl groups could be photo-dimerized thus the copolymer chains were linked inter-/intra-molecularly. Consequently, the area the copolymer chains could stabilize would decrease, leading to the demulsification. Reductive-responsive emulsions were prepared using a proteinoid sulfide as an emulsifier [9]. When exposed to a reduction condition, the disulfide bond was broken down to form a thiol proteinoid whose interfacial activity was not high enough to stabilize the oil droplets. Recently, NIR-responsive emulsion was prepared by emulsifying oil in gold nanoparticle (GNP) solution using a thermosensitive amphiphilic copolymer (poly(2-hydroxyethyl acrylate-*co*-propyl methacrylate)) as an emulsifier [10]. Upon NIR irradiation, heat was generated by the surface plasmon resonance of GNP and the emulsion underwent coalescence and phase separation possibly due to the thermal contraction of the copolymer chains. In this study, oxidation-responsive emulsion was prepared by stabilizing mineral oil droplets using an oxidizable amphiphilic copolymer (i.e., poly(vinyl pyrrolidone-*co*-allyl phenyl sulfide), P(VP-APS)). When exposed to an oxidizing condition, the sulfide of APS would be oxidized to sulfoxide and sulfone. As a result, the amphiphilicity and the interfacial activity of the copolymer would be weakened and the oil droplets-stabilizing capacity would also fall down, leading to the coalescence of oil droplets and the phase separation of emulsions (Figure 1). P(VP-APS) of different APS contents were prepared by a free radical reaction. The H_2_O_2_-casued oxidation of APS was confirmed by ^1^H nuclear magnetic resonance (NMR) and X-ray Photoelectron Spectroscopy (XPS). The effect of the oxidation of APS on the amphiphilicity and the interface activity of the copolymers were examined by UV spectroscopy and tensiometer study. The oxidation-induced destabilization of the emulsions was investigated in terms of coalescence (size increment) and phase separation.

## 2. Materials and Methods

### 2.1. Materials

Vinyl pyrrolidone (VP) and mineral oil were purchased from Sigma-Aldrich Chemical Co. (St. Louis, MO, USA). Allyl phenyl sulfide (APS) was purchased from Tokyo Chemical Industry Co. Ltd., (Tokyo, Japan). 2,2’-Azobisisobutyronitrile (AIBN) was purchased from Junsei Chemical Co. (Tokyo, Japan). Tetrahydrofuran, hexane and H_2_O_2_ solution (30%) were purchased from Dae Jung Chemicals & Metals Co., Ltd. (Siheung, South Korea).

### 2.2. Preparation of Poly(Vinyl Pyrrolidone-co-allyl Phenyl Sulfide)

Poly(vinyl pyrrolidone-*co*-allyl phenyl sulfide) (P(VP-APS)) was prepared by a free radical reaction. Each of monomers (VP and APS) was dissolved in 50 mL of tetrahydrofuran contained in a 250 mL round bottom flask so that the VP/APS molar ratio was varied (i.e., 100:0, 98:2, 97.5:2.5 and 96:4) while the total mass of monomers was kept constant (5.557 g). The mixture solutions were degassed using N_2_ stream for 30 min, 40 mg of AIBN was put to each of them and they were heated to 75 °C by immersing the round bottom flask in an oil bath and stirred using a magnetic bar for 12 h at the same temperature with reflux. For the precipitation of the copolymer, each of the reaction mixtures was poured into 600 mL of hexane contained in a 1 L beaker after they were cooled to a room temperature. The precipitate was filtered through a filter paper (Whatman no. 2) and obtained as a cake. For the purification, the cake was dissolved in the good solvent (tetrahydrofuran) and re-precipitated in the non-solvent (hexane). The cake was dried in a vacuum oven thermostated at 50 °C. The copolymer obtained from the reaction mixture whose the VP/APS molar ratio was a/b was abbreviated to P(VP-APS)(a/b).

### 2.3. Oxidizing Treatment of Poly(Vinyl Pyrrolidone-co-allyl Phenyl Sulfide)

0.2 g each of copolymers was dissolved in 10 mL of H_2_O_2_ solution (10%, in distilled water) contained in a 20 mL vial and it was stood at room temperature for 3 days. The solution was freeze-dried for the characterization. P(VP-APS)(a/b) subjected to the oxidizing treatment was termed as Oxi-P(VP-APS)(a/b).

### 2.4. ^1^H NMR Spectroscopy

P(VP-APS)(a/b) and Oxi-P(VP-APS)(a/b) were further dried overnight with a drying agent (i.e., phosphorus pentoxide) in a vacuum oven (50 °C). Each copolymer was dissolved in DMSO-d_6_ so that the concentration was 12 mg/mL. The copolymer solution was put in an NMR tube (0.5 mm outer diameter and 180 mm length), sealed tightly and it was subjected to ^1^H NMR spectroscopy on a Fourier Transform (FT) NMR Spectrometer (Bruker DPX 400 MHz (9.4T), Bruker Analytik GmbH, Karlsruhe, Germany, installed in the Central Laboratory Center of Kangwon National University). The sweep width was 12,335.526 Hz, the acquisition time was 1.3282462 s and the recycle delay was 1.00 s.

### 2.5. X-ray Photoelectron Spectroscopy (XPS)

XPS analysis was performed on P(VP-APS)(100/0), Oxi-P(VP-APS)(100/0), P(VP-APS)(96/4) and Oxi-P(VP-APS)(96/4) to investigate if the sulfide copolymer was oxidized by H_2_O_2_ treatment. A 180° double focusing hemispherical analyzer equipped with 128-channel position sensitive detector was used to detect the binding energy of atomic electrons. The beam irradiated to the copolymers was A MaKa (1253.6 eV) achromatic X-ray source. The XPS spectrum was taken at room temperature under 5 × 10^−8^ mbar or less. The binding energy of other atomic electrons was determined using the observed binding energy of C 1s electron (284.5 eV) as the reference value.

### 2.6. Temperature-Dependent Optical Density Change of Copolymer Solutions

Sixty mg each of P(VP-APS)(a/b) and Oxi-P(VP-APS)(a/b) was dissolved in 3 mL of distilled water contained in a 10 mL glass vial so that the concentration was 2% (w/v). 0.8 mL each of the copolymer solutions was put in a 1 mL cuvette and the optical density was determined at 600 nm on a UV spectrophotometer (7315 Spectrophotometer, JENWAY, Staffordshire, UK) equipped with a temperature controller (Peltier Controller, JENWAY, Staffordshire, UK) while the copolymer solution was heated at rate of 2 °C/min in 20–50 °C.

### 2.7. Measurements of Air/Water Interfacial Tensions

Twenty mg each of P(VP-APS)(a/b) and Oxi-P(VP-APS)(a/b) was dissolved in 20 mL of distilled water contained in a 30 mL glass vial so that the concentration was 1 mg/mL. The copolymer solutions underwent a serial two-times dilution until the concentration became 0.0039 mg/mL. The air/water interfacial tensions of the copolymer solutions were determined by a ring method using a tensiometer (DST 60, SEO Co., Gunpo, South Korea).

### 2.8. Preparation of Emulsion Stabilized with of Poly(Vinyl Pyrrolidone-co-allyl Phenyl Sulfide)

Each of P(VP-APS)(a/b) was dissolved in distilled water and H_2_O_2_ solution (10% (v/v), in distilled water) so that the concentration was 0.5% (w/v). 4.5 mL of aqueous solution was put in a 10 mL glass vial and 0.5 mL of mineral oil was added to the solution. The mixture was sonicated at 27% intensity for 90 s using a tip type sonicator (VC 505, Sonic & Materials, Newtown, CT, USA).

### 2.9. Investigation of Emulsion Stability

The stability of emulsion was evaluated in terms of oil droplet size. Vials containing emulsions were filled with nitrogen, tightly sealed and stored at 25 °C under dark condition for 28 days. The microphotographs of emulsions were taken at a given time and the diameter of oil droplets were determined by an image analyzer (Image-Pro Plus version 5.1, Media Cybernetics, Rockville, MD, USA). In addition, the stability of emulsion was evaluated in terms of phase separation and the following equation was used to determine the degree of phase separation [11,12].
(1)$$\mathrm{Stability}\;\left(\%\right)=\frac{\mathrm{Total}\;\mathrm{height}-\left(\mathrm{Height}\;\mathrm{of}\;\mathrm{water}\;\mathrm{layer}+\mathrm{Height}\;\mathrm{of}\;\mathrm{oil}\;\mathrm{layer}\right)}{\mathrm{Total}\;\mathrm{height}}\times 100.$$

## 3. Results and Discussion

### 3.1. ^1^H NMR Spectroscopy

Figure 2A shows the ^1^H NMR spectrum of P(VP-APS)(100/0). The methylene of vinyl group was found at 1.3 ppm, the methine at 3.7 ppm, two adjacent methylenes next to the carbonyl group of pyrrolidone group at 1.8 ppm and 2.1 ppm and the methylene next to the nitrogen atom at 3.2 ppm. Figure 2B shows the ^1^H NMR spectrum of Oxi-P(VP-APS)(100/0). The proton signals appeared at the same position as those of P(VP-APS)(100/0). This suggested that the homopolymer (i.e., P(VP-APS)(100/0)) was stable against the oxidizing agent (H_2_O_2_). In fact, no oxidizable groups were contained in the structure of P(VP-APS)(100/0). Figure 2C shows the ^1^H NMR spectrum of P(VP-APS)(98/2). Regarding the signals of the APS, the methylene of the vinyl group was found at 1.6 ppm, the methine at 1.9 ppm, the methylene next to sulfur atom at 2.4 ppm and the phenyl protons in 7.3–7.6 ppm. The VP signals of P(VP-APS)(98/2) were found at the same position as those of P(VP-APS)(100/0), indicating that VP could be copolymerized with APS without chemical deterioration. Using the total signal area of methine and methylene next to the nitrogen atom of VP and the signal area of aromatic proton at 4 position of APS, the VP/APS molar ratio was calculated to be 100:3.4. Since the VP/APS molar ratio in the reaction mixture was 98:2, it could be said that the APS was more reactive than VP. Figure 2D shows the ^1^H NMR spectrum of Oxi-P(VP-APS)(98/2). The VP signals of Oxi-P(VP-APS)(98/2) were found at the same position as those of P(VP-APS)(98/2). However, the signals of the phenyl group and the methylene next to sulfur atom of Oxi-P(VP-APS)(98/2) appeared in more downfield than those of P(VP-APS)(98/2). The signals of the phenyl group of Oxi-P(VP-APS)(98/2) were found in 7.6–7.7 ppm and those of P(VP-APS)(98/2) in 7.3–7.6 ppm. Sulfide is known to be oxidized to sulfoxide and sulfone under an oxidative condition. The APS of P(VP-APS)(98/2) was likely to be oxidized by the oxidizing agent (i.e., H_2_O_2_). Upon the oxidation, oxygen (an electronegativity atom) is attached to the sulfur, the neighboring hydrogen atoms would decrease in their electron density and be deshielded. This could account for why the signal of the phenyl group and that of the methylene next to sulfur shifted to downfield. P(VP-APS)(97.5/2.5) and P(VP-APS)(96/4) exhibited their signals at the same position as P(VP-APS)(98/2) (Appendix A). Three kinds of copolymers were composed of the same monomers. However, the compositional ratio would be different because they were prepared from the reaction mixtures that differed in VP/APS molar ratio. As described previously, the VP/APS molar ratio could be calculated using the signal area of proton belonging to VP and that of proton belonging to APS. The VP/APS molar ratio of P(VP-APS)(97.5/2.5) and P(VP-APS)(96/4) were found to be 100:3.6 and 100:4.8, respectively. Like P(VP-APS)(98/2), P(VP-APS)(97.5/2.5) and P(VP-APS)(96/4) showed higher VP/APS molar ratios than their corresponding reaction mixtures, reconfirming that the reactivity of ASP was higher than that of VP. Like P(VP-APS)(98/2), P(VP-APS)(97.5/2.5) and P(VP-APS)(96/4) exhibited the signals of phenyl group and methylene, which shifted to downfield after oxidized by H_2_O_2_ (Appendix A). The oxidation-induced deshielding of the neighboring protons would be responsible for the chemical shift of the phenyl protons and the methylene ones.

### 3.2. X-ray Photoelectron Spectroscopy

Figure 3A shows the XPS spectrum of P(VP-APS)(100/0) and Oxi-P(VP-APS)(100/0). No significant signals were found in the spectrum of P(VP-APS)(100/0) and in that of Oxi-P(VP-APS)(100/0) either. The binding energy window scanned is the range where the signal of sulfur atom appears. In fact, no sulfur was in the structure of P(VP-APS)(100/0), thus no sulfur would be in the structure of Oxi-P(VP-APS)(100/0) either. Figure 3B shows the XPS spectrum of P(VP-APS)(96/4) and Oxi-P(VP-APS)(96/4). In the spectrum of P(VP-APS)(96/4), a peak was found around 163 eV and it could be assigned to the sulfide of APS. The binding energy of sulfur electron of sulfide was reported to be 163–164 eV [13]. In the spectrum of Oxi-P(VP-APS)(96/4), a sharp peak appeared around 168 eV and it could be attributed to the sulfone of Oxi-P(VP-APS)(96/4). The binding energy of sulfur electron of sulfide was reported to be 168 eV [13]. Sulfide compounds are known to be oxidized to sulfoxides and further to sulfones. Since the sulfone signal was found on the XPS spectrum of Oxi-P(VP-APS)(96/4), it could be said that the ally phenyl sulfide of P(VP-APS)(96/4) was oxidized to ally phenyl sulfone under present oxidative condition.

### 3.3. Temperature-Dependent Optical Density Change of Copolymer Solutions

Figure 4 shows the temperature-dependent optical density change of copolymer solutions (P(VP-APS)(100/0), Oxi-P(VP-APS)(100/0), P(VP-APS)(98/2), Oxi-P(VP-APS)(98/2), P(VP-APS)(97.5/2.5), Oxi-P(VP-APS)(97.5/2.5), P(VP-APS)(96/4) and Oxi-P(VP-APS)(96/4)). It was reported that the copolymerization of VP with a hydrophobic monomer led to the formation of a polymer exhibiting a lower critical solution temperature (LCST) [14]. APS is a kind of hydrophobic monomer thus P(VP-APS)(a/b) and Oxi-P(VP-APS)(a/b) would be able to show LCST behavior. This was the reason why the temperature-dependent optical density change of copolymer solutions was observed. However, all the copolymer solutions kept their optical density almost constant in the temperature range tested, suggesting that none of them showed LCST behavior. The content of APS in the copolymers might be too low or high to exhibit LCST behavior in the temperature window but it is not clear yet. Interestingly, the optical density was strongly dependent on the APS content. For example, the optical density at 20 °C of P(VP-APS)(100/0), P(VP-APS)(98/2), P(VP-APS)(97.5/2.5) and P(VP-APS)(96/4) solution was 0.01, 0.23, 0.38 and 0.5, respectively. The optical density increased with increasing the APS content of the copolymers. Since VP is a hydrophilic monomer and APS is a lipophilic monomer, P(VP-APS)(a/b) except for P(VP-APS)(100/0) was likely to be amphiphilic. Amphiphilic molecules are able to be assembled into self-assemblies in aqueous solution by an entropy-driven process [15,16,17,18,19]. The copolymers were thought to be assembled into polymeric micelles. Air/water interfacial tension profiles of copolymer solutions shown in Figure 5 would also be an evidence for the presence of the micelles. A L-type of interfacial tension profile is typical of a surface-active agent and an indication of micelle formation [20,21]. The significantly appreciable value of the optical density of the copolymer solutions could be ascribed to the formation of a self-assembly. Since APS can endow the copolymer with amphiphilic property, the amphiphilicity of the copolymer and the degree of self-assembling would increase as the content of the hydrophobic monomer increases. This could account for why the optical density increased with increasing the APS content. P(VP-APS)(100/0), a homo polymer, would not be amphiphilic and hardly be assembled in aqueous solution, resulting in almost zero optical density. On the other hand, the optical density of Oxi-P(VP-APS)(a/b) solution, except for Oxi-P(VP-APS)(100/0) solution, was lower than that of its corresponding P(VP-APS)(a/b) solution. For example, the optical density of Oxi-P(VP-APS)(98/2), Oxi-P(VP-APS)(97.5/2.5) and Oxi-P(VP-APS)(96/4)) were 0.07, 0.11 and 0.34, respectively. As described previously, sulfides can be oxidized to sulfoxide and sulfones under an oxidative condition. In fact, the APS of a copolymer (i.e., P(VP-APS)(96/4)) was oxidized to the sulfone via H_2_O_2_ treatment (Figure 3). Upon the oxidation of APS, it would become hydrophilic, thus the amphiphilicity of the copolymer and the degree of self-assembling would decrease. This would be a reason why the optical density of the copolymer solutions decreased via H_2_O_2_ treatment.

### 3.4. Measurements of Air/Water Interfacial Tensions

Figure 5A shows the air/water interfacial tension profile of P(VP-APS)(100/0) solution and Oxi-P(VP-APS)(100/0) solution. The interfacial tension of P(VP-APS)(100/0) solution slightly decreased from 72 to 70 dyne/cm in the full range of concentration tested (0 to 1.0 mg/mL). The interfacial tension profile of Oxi-P(VP-APS)(100/0) solution was almost the same as that of P(VP-APS(100/0), suggesting that the interfacial activity of P(VP-APS)(100/0) was little affected by H_2_O_2_ treatment. Figure 5B shows the air/water interfacial tension profile of P(VP-APS)(98/2) solution and Oxi-P(VP-APS)(98/2) solution. The interfacial tension of P(VP-APS)(98/2) solution markedly decreased from 72.5 to 64.5 dyne/cm in 0 to 0.25 mg/mL and it was almost constant in the remaining concentration range (0.25–10 mg/mL). A L-type of interfacial tension profile is typical of a surface-active agent and an indication of micelle formation [20,21]. Since the plateau interfacial tension of P(VP-APS)(98/2) solution was less than that of P(VP-APS)(100/0) solution, it was concluded that the interfacial activity of the copolymer was higher than that of the homopolymer. As described previously, VP is a hydrophilic monomer and APS is a hydrophobic one, the copolymerization of VP with APS would result in the formation of an amphiphilic and surface-active copolymer. This could explain why P(VP-APS)(98/2) was more surface-active than P(VP-APS)(100/0). The interfacial tension profile of Oxi-P(VP-APS)(98/2) solution resembled that of P(VP-APS)(98/2) solution. However, the plateau interfacial tension, about 67.1 dyne/cm, was significantly higher than that of P(VP-APS)(98/2) solution. That is, the interfacial activity of the H_2_O_2_-treated copolymer (i.e., Oxi-P(VP-APS)(98/2)) was less than the untreated copolymer (i.e., P(VP-APS)(98/2)). As shown in Figure 3, the sulfide of APS of a copolymer composed of VP and APS was oxidized to the sulfone by H_2_O_2_ treatment. Accordingly, the lipophilicity of APS would decrease, leading to a decrease in the amphiphilicity of the copolymer. This would be a reason the interfacial activity of P(VP-APS)(98/2) decreased by H_2_O_2_ treatment. Figure 5C shows the air/water interfacial tension profile of P(VP-APS)(97.5/2.5) solution and Oxi-P(VP-APS)(97.5/2.5) solution. The interfacial tension of P(VP-APS)(97.5/2.5) solution also decreased in a saturation manner. The plateau interfacial tension was about 62.3 dyne/cm and it was significantly lower than that of P(VP-APS)(98/2) solution, suggesting that the interfacial activity of P(VP-APS)(97.5/2.5) was higher than that of P(VP-APS)(98/2). This was possibly because the APS content of P(VP-APS)(97.5/2.5) (3.47%) was higher than that of P(VP-APS)(98/2) (3.29%) thus the amphiphilicity of the former copolymer would be stronger than that of the latter one. Like P(VP-APS)(98/2) solution, P(VP-APS)(97.5/2.5) solution showed an increased interfacial tension (i.e., a decreased interfacial activity) after the copolymer was treated with H_2_O_2_. For example, the plateau interfacial tension of Oxi-P(VP-APS)(97.5/2.5) was 63.6 dyne/cm and it was significantly higher than that of P(VP-APS)(97.5/2.5), 62.3 dyne/cm. The oxidation of APS would be responsible for the decreased interfacial activity. Figure 5D shows the air/water interfacial tension profile of P(VP-APS)(96/4) solution and Oxi-P(VP-APS)(96/4) solution. Like the other copolymer solutions, P(VP-APS)(96/4) solution exhibited a L-type of interfacial tension profile. P(VP-APS)(96/4) solution showed the lowest interfacial tensions (the highest interfacial activity) among the copolymer solutions tested. P(VP-APS)(96/4) had the highest APS content (4.8%) thus its amphiphilicity would be the strongest, accounting for the highest interfacial activity. Like the other copolymer solutions, P(VP-APS)(96/4) solution showed an increased interfacial tension (i.e., a decreased interfacial activity) after treated with H_2_O_2_. For example, the minimum interfacial tension of Oxi-P(VP-APS)(96/4) was 61.5 dyne/cm and it was significantly higher than that of P(VP-APS)(96/4), 57.0 dyne/cm. The decreased interfacial activity was possibly due to the oxidation of APS (Figure 3).

### 3.5. Investigation of Emulsion Stability

Figure 6A shows the time-dependent droplet diameter of O/W emulsions stabilized with P(VP-APS)(100/0), P(VP-APS)(98/2), P(VP-APS)(97.5/2.5) and P(VP-APS)(96/4) when distilled water were used as the aqueous phase. The droplet diameter of emulsions stabilized with P(VP-APS)(100/0) increased fast from 2 to 55 μm for 24 h and thereafter droplets could hardly be found. The fast increase in diameter indicated that the oil droplets underwent coalescence for such a short period. P(VP-APS)(100/0) is the homopolymer of vinyl pyrrolidone and its amphiphilic property would be weak. In fact, the interfacial activity of P(VP-APS)(100/0) was very low (Figure 5). Thus, the homopolymer would hardly be able to reduce the O/W interfacial energy and to stabilize the oil droplets. The droplet diameter of emulsion stabilized with P(VP-APS)(98/2) increased much more slowly than that of emulsion stabilized with P(VP-APS)(100/0). For examples, the droplet diameter of emulsion stabilized with P(VP-APS)(98/2) increased slowly from 4 to 31 μm for 336 h. The interfacial activity of P(VP-APS)(98/2) was significantly higher than that of P(VP-APS)(100/0) (Figure 5). Thus, P(VP-APS)(98/2) would be able to decrease the air/water interfacial energy and to stabilize the droplets more effectively. On the other hand, the droplet diameter of emulsion stabilized with P(VP-APS)(97.5/2.5) and with P(VP-APS)(96/4) decreased from 3 to 28 μm and 3 to 16 μm, respectively. Since the increase rate of droplet size was higher in the order of emulsion stabilized with P(VP-APS)(100/0) > with P(VP-APS)(98/2) > with P(VP-APS)(97.5/2.5) > with P(VP-APS)(96/4), it could be said that the droplet-stabilizing efficacy was greater in the reversed order (P(VP-APS)(94/4) > P(VP-APS)(97.5/2.5) > P(VP-APS)(98/2) > P(VP-APS)(100/0)). In general, the droplet-stabilizing efficacy of a surfactant is higher as the interfacial activity is higher. In fact, the interfacial activity was greater in the order of P(VP-APS)(96/4) > P(VP-APS)(97.5/2.5) > P(VP-APS)(98/2) > P(VP-APS)(100/0) (Figure 5). Figure 6B shows the time-dependent droplet diameter of O/W emulsions stabilized with P(VP-APS)(100/0), P(VP-APS)(98/2), P(VP-APS)(97.5/2.5) and P(VP-APS)(96/4) when H_2_O_2_ solution was used as the aqueous phase. Like when distilled water was used as aqueous phase, the droplet diameter of emulsions stabilized with P(VP-APS)(100/0) increased fast from 2 to 51 μm for 24 h due to the poor stabilizing efficacy of P(VP-APS)(100/0). The droplet diameter of emulsion stabilized with P(VP-APS)(98/2) increased slowly from 2 to 16 μm for the first 170 h then increased rapidly up to 73 μm for the remaining period. The droplet diameter of emulsion stabilized with P(VP-APS)(97.5/2.5) increased in a similar manner and the increase profile was almost the same as that of the droplet diameter of emulsion stabilized with P(VP-APS)(98/2). The droplet diameter of emulsion stabilized with P(VP-APS)(96/4) also increased in a similar pattern but the increase rate was lower than that of the droplet diameter of emulsion stabilized with P(VP-APS)(98/2) and with P(VP-APS)(97.5/2.5). This was possibly because the former copolymer had a higher APS content and it showed a stronger interfacial activity (Figure 5). In the XPS spectra of P(VP-APS)(96/4) and Oxi-P(VP-APS)(96/4), it seemed that the sulfide of APS could be completely transformed into the sulfone 3 days after H_2_O_2_ treatment (Figure 3). However, the stability of droplet diameter of O/W emulsions stabilized with P(VP-APS)(98/2), P(VP-APS)(97.5/2.5) or P(VP-APS)(96/4) in H_2_O_2_ solution was not markedly different from that in water during 200 h incubation. VP is a hydrophilic monomer and APS is a hydrophobic one, the copolymerization of VP with APS would result in the formation of an amphiphilic and surface-active copolymer. In O/W emulsion, APS was likely to orient toward the oil droplet of the emulsion. Thus, it would be less exposed to H_2_O_2_ contained in the water phase of emulsion than to the oxidizing agent in aqueous solution, which was used to oxidize the copolymer for the XPS spectroscopy. That is, the oxidization of APS would be able to be delayed because APS can be sequestered by oil phase. This could explain why there was no big difference in the stability of droplet diameter between the emulsions in water and those in H_2_O_2_ solution during 200 h incubation. However, the stability of droplet diameter of O/W emulsions in H_2_O_2_ solution was much less than that of emulsions in water when the incubation period was longer than 200 h (Figure 6). Upon H_2_O_2_ treatment, the sulfoxide of the APS of a copolymer was oxidized to the sulfone (Figure 3) and the interfacial activity of the copolymer significantly decreased (Figure 5). Therefore, the copolymer contained in emulsion prepared using H_2_O_2_ solution would be able to be oxidized and its interfacial activity would be lower than that of the copolymer contained in emulsion prepared using distilled water. This could explain why emulsions prepared using H_2_O_2_ solution were more unstable than emulsion prepared using distilled water in terms of droplet size. Figure 7A shows the stability of O/W emulsions stabilized with P(VP-APS)(100/0), P(VP-APS)(98/2), P(VP-APS)(97.5/2.5) and P(VP-APS)(96/4) when distilled water was used as the aqueous phase. The stability of emulsion stabilized with P(VP-APS)(100/0) fell down very fast to almost zero in 48 h due to the poor interfacial activity of the homopolymer. Since P(VP-APS)(100/0) was not capable of stabilizing the oil droplets, the coalescence of the droplets would readily take place, leading to a fast phase separation (i.e., a fast decrease in the stability). The stability of emulsion stabilized with P(VP-APS)(98/2), P(VP-APS)(97.5/2.5) and P(VP-APS)(96/4) slowly decreased and reached zero in 650 h. P(VP-APS)(a/b) was interface-active (Figure 5) and the copolymers could stabilize oil droplet more effectively than the homopolymer (Figure 6). This could account for why the stability of emulsion stabilized with the copolymers was higher than that of emulsion stabilized with the homopolymer. Considering the slope of curves, it was concluded that the stability was greater in the order of emulsion stabilized with P(VP-APS)(94/4) > with P(VP-APS)(97.5/2.5) > with P(VP-APS)(98/2) > with P(VP-APS)(100/0). The order of emulsion stability against the phase separation (Figure 7A) was the same as that of oil droplet stability against the coalescence (Figure 6A). Figure 7B shows the stability of O/W emulsions stabilized with P(VP-APS)(100/0), P(VP-APS)(98/2), P(VP-APS)(97.5/2.5) and P(VP-APS)(96/4) when H_2_O_2_ solution was used as the aqueous phase. The stability of emulsions decreased in a similar pattern to that of emulsions prepared using distilled water as the aqueous phase. However, the stability decreased more rapidly than that of emulsions prepared using distilled water. For example, the stability of P(VP-APS)(98/2)-stabilized emulsion prepared using distilled water was 45% at the time lapse of 350 h whereas the stability of the emulsion prepared using H_2_O_2_ solution was almost 0% at the same time lapse. As described previously, the sulfide of APS of P(VP-APS)(a/b) contained in the emulsion prepared using H_2_O_2_ solution would be able to be oxidized to the sulfone and the interfacial activity and the droplet-stabilizing capability of the copolymers would decrease, leading to a droplet coalescence and the phase separation.

## 4. Conclusions

Oxidation-responsive emulsions were prepared using oxidizable copolymers (i.e., P(VP-APS)(a/b)) as an emulsifier. ^1^H NMR and XPS spectroscopy revealed that the sulfide of APS was oxidized by H_2_O_2_ treatment. Through UV spectroscopy and tensiometer study, the amphiphilicity and the interface activity of the copolymers were found to be markedly weakened by the oxidizing agent. The destabilization of emulsions stabilized with P(VP-APS)(a/b) was expedited when H_2_O_2_ solution (10%, v/v) was used as an aqueous phase, possibly due to the oxidation of APS and the weakened interfacial activity. P(VP-APS)(a/b) prepared in present study would be able to be used as an oxidizable emulsifier for the preparation of O/W emulsions that are required to readily be demulsified on demand.

## Figures and Tables

**Figure 1 polymers-12-00498-f001:**
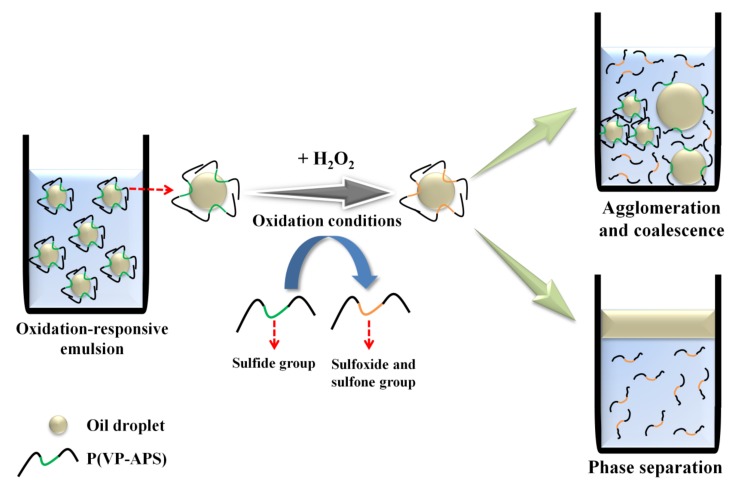
Schematic representation of oxidation-responsive emulsion. Oxidation-responsive emulsion was prepared by stabilizing mineral oil droplets using an oxidizable amphiphilic copolymer (i.e., poly(vinyl pyrrolidone-co-allyl phenyl sulfide) (P(VP-APS)). When exposed to an oxidizing condition, the sulfide of APS would be oxidized to sulfoxide and sulfone. As a result, the amphiphilicity and the interfacial activity of the copolymer would be weakened and the oil droplets-stabilizing capacity would also fall down, leading to the coalescence of oil droplets and the phase separation of emulsions.

**Figure 2 polymers-12-00498-f002:**
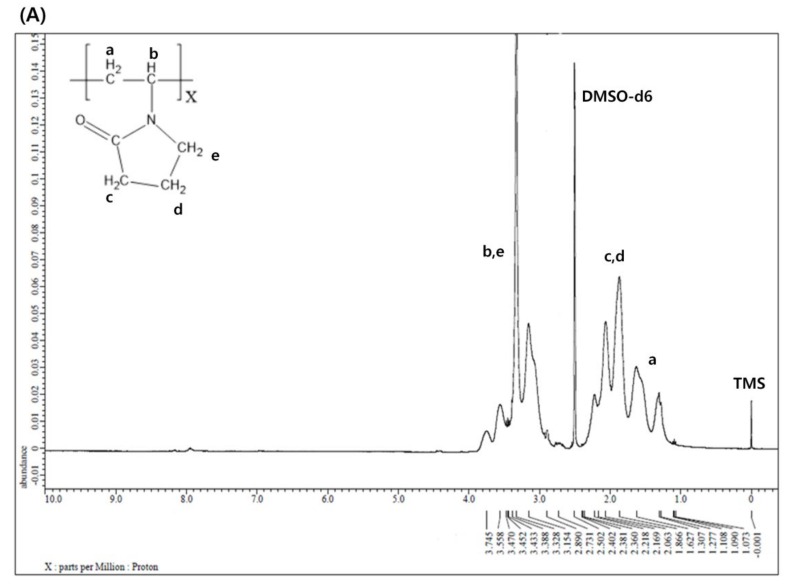
^1^H nuclear magnetic resonance (NMR) spectrum of P(VP-APS)(100/0) (**A**), Oxi-P(VP-APS)(100/0) (**B**), P(VP-APS)(98/2) (**C**) and Oxi-P(VP-APS)(98/2) (**D**). Inset represents the enlarged signals in 7 to 7.8 ppm.

**Figure 3 polymers-12-00498-f003:**
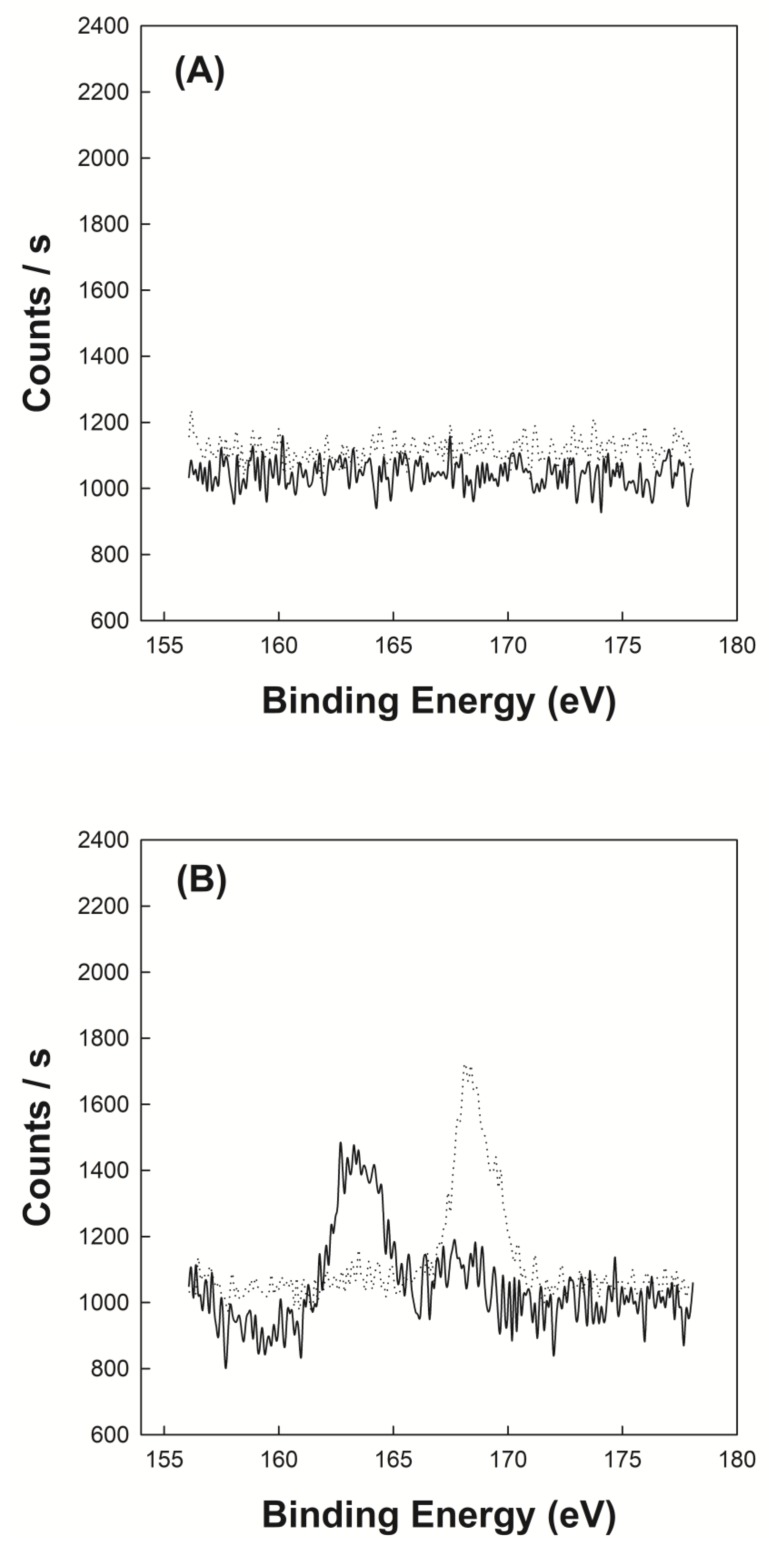
XPS spectrum of P(VP-APS)(100/0) (**A**, solid line), Oxi-P(VP-APS)(100/0) (**A**, dotted line), P(VP-APS)(96/4) (**B**, solid line) and Oxi-P(VP-APS)(96/4) (**B**, dotted line).

**Figure 4 polymers-12-00498-f004:**
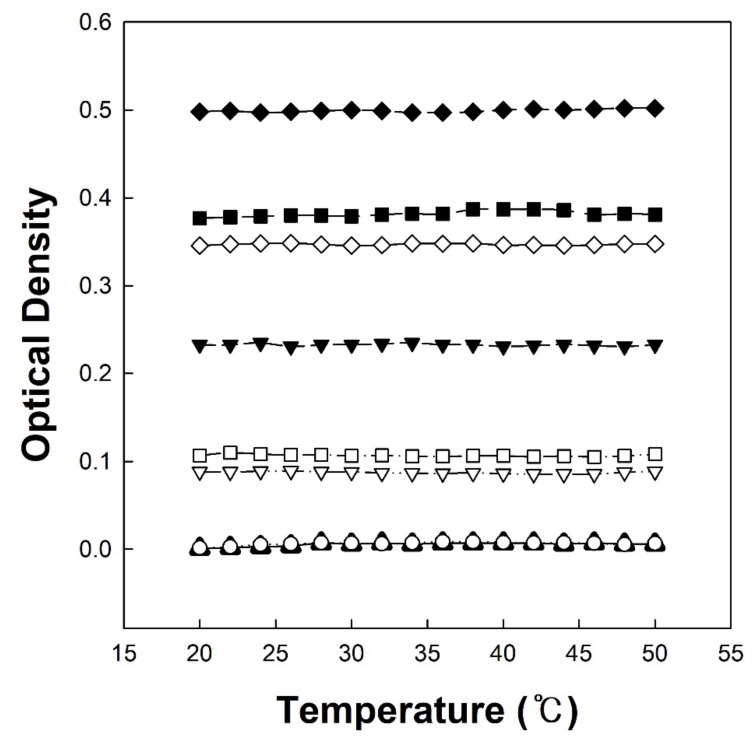
Temperature-dependent optical density change of copolymer solutions (P(VP-APS)(100/0) (▲), Oxi-P(VP-APS)(100/0) (○), P(VP-APS)(98/2) (▼), Oxi-P(VP-APS)(98/2) (▽), P(VP-APS)(97.5/2.5) (■), Oxi-P(VP-APS)(97.5/2.5) (□), P(VP-APS)(96/4) (◆) and Oxi-P(VP-APS)(96/4) (◇)).

**Figure 5 polymers-12-00498-f005:**
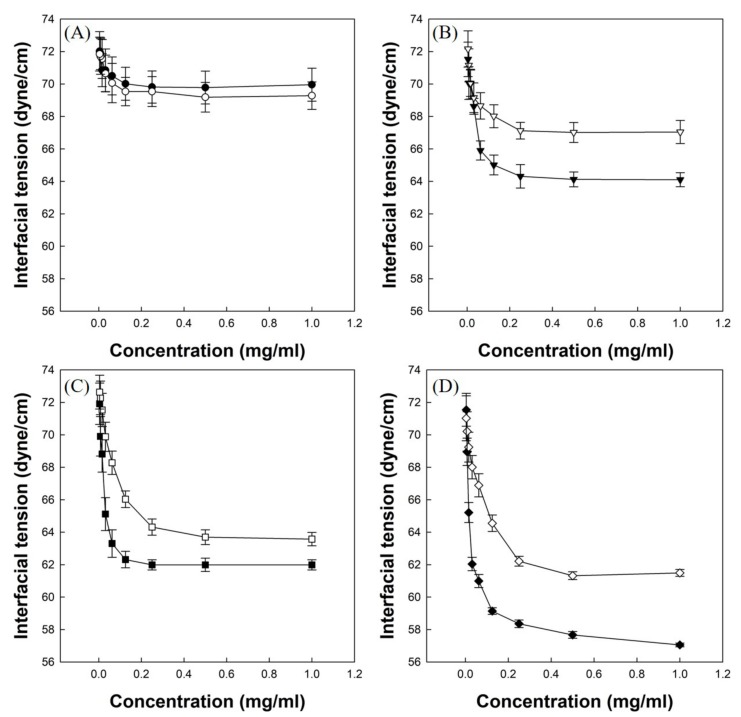
Air/water interfacial tension profiles of copolymer solutions ((P(VP-APS)(100/0) (●) and Oxi-P(VP-APS)(100/0) (○) (**A**), (P(VP-APS)(98/2) (▼) and Oxi-P(VP-APS)(98/2) (▽) (**B**), P(VP-APS)(97.5/2.5) (■) and Oxi-P(VP-APS)(97.5/2.5) (□) (**C**), P(VP-APS)(96/4) (◆) and Oxi-P(VP-APS)(96/4) (◇) (**D**)).

**Figure 6 polymers-12-00498-f006:**
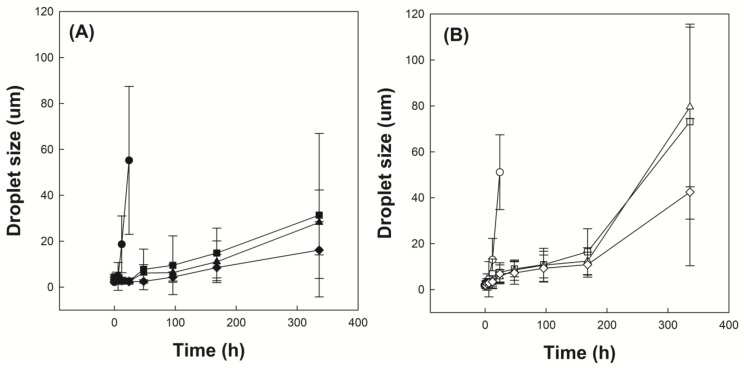
Time-dependent droplet diameter of O/W emulsions stabilized with P(VP-APS)(100/0) (●,○), P(VP-APS)(98/2) (▲,△), P(VP-APS)(97.5/2.5) (■,□) and P(VP-APS)(96/4) (◆,◇), when distilled water (**A**) and H_2_O_2_ solution (**B**) were used as the aqueous phase.

**Figure 7 polymers-12-00498-f007:**
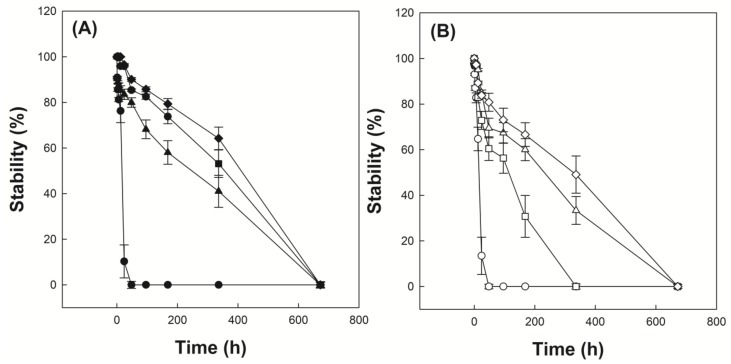
Stability of O/W emulsions stabilized with P(VP-APS)(100/0) (●,○), P(VP-APS)(98/2) (▲,△), P(VP-APS)(97.5/2.5) (■,□) and P(VP-APS)(96/4) (◆,◇), when distilled water (**A**) and H_2_O_2_ solution (**B**) were used as the aqueous phase.

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
