# Peer review of "Oxidation–Responsive Emulsions Stabilized with Poly(Vinyl Pyrrolidone-co-allyl Phenyl Sulfide)"

_polymers, 2020, doi:10.3390/polym12020498_

Round 1
Reviewer 1 Report
Overall:
This paper presents synthesis and characterization of poly(vinyl pyrrolidone‐co‐allyl phenyl sulfide) (P(VP‐APS)) in which the sulfide of APS can be oxidized by H2 O2 treatment. The authors explored the effect of oxidation of APS on the amphiphilicity and the interface activity of the copolymers. They also wanted to use the oxidation-responsive property of (P(VP‐APS) as an emulsifier for oil to obtain readily be demulsified on demand. A reasonable amount of characterization are presented. However, the manuscript was poor written and suffers from some grammatical mistakes and typos which need a strong polishing. The abstract and introduction are too simple and should be rewritten and reorganized in order to the readers have a good understanding of this work. For the reason stated above, my recommendation is not to publish the paper in its current form.
Specific comments
In Figure 5, the author should divide the graph as Figure A, B, C and D, to make it easier for readers to understand. For Figure 6B and 7B, it should be pointed out what do these symbols represent? In the XPS spectra of P(VP‐APS) (96/4) and Oxi‐P(VP‐APS) (96/4), it seems that the sulfide of APS can be completely transformed into the sulfone after 3days H2O2 treatment. However, the stability of droplet diameter of O/W emulsions stabilized with P(VP‐APS)(97.5/2.5) or P(VP‐APS)(96/4) in water is almost similar as that in H2O2 solution during 200h incubation. If the droplet coalescence and the phase separation are really caused by the oxidization of sulfide of APS? why did not it show a rapid oxization-responsive demulsification?Author Response
To reviewer 1
Thank you for your kind and critical comments.
The following are our itemized responses to your comments.
Overall:
This paper presents synthesis and characterization of poly(vinyl pyrrolidone‐co‐allyl phenyl sulfide) (P(VP‐APS)) in which the sulfide of APS can be oxidized by H2 O2 treatment. The authors explored the effect of oxidation of APS on the amphiphilicity and the interface activity of the copolymers. They also wanted to use the oxidation-responsive property of (P(VP‐APS) as an emulsifier for oil to obtain readily be demulsified on demand. A reasonable amount of characterization are presented. However, the manuscript was poor written and suffers from some grammatical mistakes and typos which need a strong polishing. The abstract and introduction are too simple and should be rewritten and reorganized in order to the readers have a good understanding of this work. For the reason stated above, my recommendation is not to publish the paper in its current form.
Response:
A native speaker has got through with the manuscript carefully and correct some grammatical mistakes and typos (Please see the revised manuscript). The Abstract has been rewritten so that it is be informative. The introduction has been somewhat reorganized so that the readers can easily understand the present work.
Specific comments
In Figure 5, the author should divide the graph as Figure A, B, C and D, to make it easier for readers to understand.
Response: Figure 5 has been divided into Figure 5A, B, C, and D.
For Figure 6B and 7B, it should be pointed out what do these symbols represent?
Response: The symbols in Figure 6B and Figure 7B have been defined in their captions, respectively.
In the XPS spectra of P(VP‐APS) (96/4) and Oxi‐P(VP‐APS) (96/4), it seems that the sulfide of APS can be completely transformed into the sulfone after 3days H2O2 treatment. However, the stability of droplet diameter of O/W emulsions stabilized with P(VP‐APS)(97.5/2.5) or P(VP‐APS)(96/4) in water is almost similar as that in H2O2 solution during 200h incubation. If the droplet coalescence and the phase separation are really caused by the oxidization of sulfide of APS? why did not it show a rapid oxization-responsive demulsification?
Response: VP is a hydrophilic monomer and APS is a hydrophobic one, the copolymerization of VP with APS would result in the formation of an amphiphilic and surface-active copolymer. In O/W emulsion, APS is likely to orient toward the oil droplet of the emulsion. Thus, it would be less exposed to H2O2 contained in the water phase of emulsion than to the oxidizing agent in aqueous solution, which was used to oxidize the copolymer for the XPS spectroscopy. That is, the oxidization of APS would be able to be delayed because APS can be sequestered by oil phase. This could explain why the stability of droplet diameter of O/W emulsions stabilized with P(VP‐APS)(97.5/2.5) or P(VP‐APS)(96/4) in water is almost similar as that in H2O2 solution during 200h incubation. However, the stability of droplet diameter of O/W emulsions in H2O2 solution was much less than that of emulsion in water after 200 h incubation (Please see Figure 6)
Thank you again for your kind and critical comments.
We hope our responses will meet your comments.

Reviewer 2 Report
This manuscript reports on the use of an amphiphilic copolymer that consists of an oxidation-responsive component, allyl phenyl sulfide (APS), to control size and stability of mineral oil droplets. The authors concluded that oxidation of APS changes overall amphiphilicity of the copolymer, which in turn affect properties of the emulsions. The conclusions are supported by the experimental results and the manuscript is well-written. My only question is about self-assembly of the copolymer solutions. The authors mentioned that the copolymers were thought to assemble into polymeric micelles. Additional characterizations, such as small angle x-ray scattering, would be very useful to support this hypothesis. A rheology experiment may also be useful to prove the presence of the micelles.
Author Response
To reviewer 2
Thank you for your kind and critical comments.
The following are our responses to your comments.
Comments and Suggestions for Authors
This manuscript reports on the use of an amphiphilic copolymer that consists of an oxidation-responsive component, allyl phenyl sulfide (APS), to control size and stability of mineral oil droplets. The authors concluded that oxidation of APS changes overall amphiphilicity of the copolymer, which in turn affect properties of the emulsions. The conclusions are supported by the experimental results and the manuscript is well-written. My only question is about self-assembly of the copolymer solutions. The authors mentioned that the copolymers were thought to assemble into polymeric micelles. Additional characterizations, such as small angle x-ray scattering, would be very useful to support this hypothesis. A rheology experiment may also be useful to prove the presence of the micelles.
Response: I agree with you. Additional characterizations, such as small angle x-ray scattering and a rheology experiment, would be very useful to support the presence of the micelles. Air/water interfacial tension profiles of copolymer solutions shown in Figure 5 would also be an evidence for the presence of the micelles. A L-type of interfacial tension profile is typical of a surface-active agent and an indication of micelle formation [Wesslén, B. et al. Biomaterials. 1994, 15, 278-284; Maltesh, C. et al. Langmuir. 1992, 8, 1511-1513]. The sentence has been added to the manuscript. The first author left my lab to work for his company thus it hardly allows me to perform additional characterizations. You may understand the situation I am facing.
Thank you again for your kind and critical comments.
Round 2
Reviewer 1 Report
Now, this paper is publishable subject.
Author Response
To reviewer 1
Thank you for your positive comments.